# Homozygous C677T Methylenetetrahydrofolate Reductase (MTHFR) Polymorphism as a Risk Factor for Endometriosis: A Retrospective Case–Control Study

**DOI:** 10.3390/ijms242015404

**Published:** 2023-10-20

**Authors:** Giovanni Delli Carpini, Luca Giannella, Jacopo Di Giuseppe, Nina Montik, Michele Montanari, Mariasole Fichera, Daniele Crescenzi, Carolina Marzocchini, Maria Liberata Meccariello, Donato Di Biase, Arianna Vignini, Andrea Ciavattini

**Affiliations:** 1Gynecologic Section, Department of Odontostomatologic and Specialized Clinical Sciences, Università Politecnica delle Marche, 60213 Ancona, Italy; giovdellicarpini@gmail.com (G.D.C.); l.giannella@staff.univpm.it (L.G.); jacopo.digiuseppe@ospedaliriuniti.marche.ut (J.D.G.); nina.montik@ospedaliriuniti.marche.it (N.M.); michele.montanari@ospedaliriuniti.marche.it (M.M.); fichera.mariasole@gmail.com (M.F.); daniele.crescenzi@ospedaliriuniti.marche.it (D.C.); carolina.mead@hotmail.it (C.M.); liberamec@gmail.com (M.L.M.); thesonofmid_@libero.it (D.D.B.); 2Section of Biochemistry, Biology and Physics, Department of Odontostomatologic and Specialized Clinical Sciences, Università Politecnica delle Marche, 60126 Ancona, Italy; a.vignini@staff.univpm.it

**Keywords:** endometriosis, MTHFR, epigenetic, oxidative stress, DNA methylation, endometriosis risk factors

## Abstract

This study was conducted to evaluate the role of methylenetetrahydrofolate reductase (MTHFR) C677T homozygous polymorphism as a risk factor for endometriosis. A retrospective case–control study was conducted from January 2020 to December 2022 on all patients attending the gynecological outpatient clinic of our institution who had performed an MTHFR polymorphisms test. Patients with endometriosis were considered cases, while those without endometriosis were considered controls. The presence of an MTHFR C677T homozygous polymorphism was defined as exposure. Risk factors for endometriosis were considered confounders in a binomial logistic regression, with endometriosis diagnosis as the dependent variable. Among the 409 included patients, 106 (25.9%) cases and 303 (74.1%) controls were identified. A higher rate of MTHFR C677T homozygous polymorphism was found in patients with endometriosis (24.5% vs. 15.8%, *p* = 0.0453), with an adOR of 1.889 (95% CI 1.076–3.318, *p* = 0.0269) at the binomial logistic regression. A history of no previous pregnancy was associated with an endometriosis diagnosis (adOR 2.191, 95% CI 1.295–3.708, *p* = 0.0035). An MTHFR C677T homozygous polymorphism could be considered a risk factor for endometriosis. Epigenetic modifications may be the most important mechanism explaining the observed association through the processes of altered DNA methylation and reduced activity of antioxidant systems.

## 1. Introduction

Endometriosis is an estrogen-dependent chronic disorder affecting 10–15% of patients of childbearing age and is characterized by the displacement of endometrial tissue at ectopic locations. It is responsible for symptoms that can heavily influence a patient’s quality of life, such as chronic pelvic pain, dysmenorrhea, and deep dyspareunia, with a significant socio-economic impact [1].

The pathogenesis of endometriosis is multifactorial and includes genetic, environmental, hormonal, inflammatory, and epigenetic mechanisms [2,3,4,5,6]. Aberrant DNA methylation seems to be the most important epigenetic mechanism [7,8,9,10,11]. Aberrant DNA methylation can be amplified by oxidative stress caused by hyperhomocysteinemia [12]. Elevated homocysteine levels can be generated in most cases by incorrect eating habits (western pattern diet) or by disorders of folate metabolism caused by methylenetetrahydrofolate reductase (MTHFR) gene polymorphisms [12,13,14]. MTHFR polymorphisms may also determine increased oxidative stress, generalized inflammatory response, and epigenetic modifications via aberrant DNA methylation, regardless of homocysteine levels [14]. Aberrant DNA methylation has been reported in endometriotic tissue in genes implicated in the hormonal and inflammatory factors of endometriosis pathogenesis (estrogen and progesterone receptors ERα, Erβ, and Prβ, the aromatase CYP19, the homebox protein HOXA-10, and the cyclooxygenase COX-2) [7,8,9,10,11,14].

Moreover, there is preliminary evidence about the association between MTHFR polymorphisms, endometriosis, and endometriosis-related infertility [15,16,17]. Therefore, it is possible to hypothesize that MTHFR polymorphisms may act as one of the risk factors for endometriosis, supporting mechanisms of aberrant DNA methylation in critical genes of endometriosis pathogenesis. To test this hypothesis, we conducted this study to evaluate the association between MTHFR polymorphisms and endometriosis diagnosis after controlling for potential confounders.

## 2. Results

During the study period, 5289 patients attended the outpatient clinic of our institution. Among them, 409 (7.7%) had previously performed an MTHFR polymorphisms test for both C677T and A1298C polymorphisms at our laboratory, with complete data. The 106 (25.9%) patients who had undergone a surgical procedure in the past with an available written histopathological diagnosis of endometriosis were considered cases. The remaining 303 (74.1%) patients who never received a histopathological diagnosis of endometriosis, had no clinical or ultrasound sign of endometriosis, and did not present any endometriosis-related symptom were considered controls.

Figure 1 reports the flowchart of the study population.

The comparison of background and clinical variables between cases and controls is reported in Table 1.

Patients with endometriosis presented a lower number of previous pregnancies: 0 (0–1) vs. 1 (0–2), a higher rate of no previous pregnancy (50.9% vs. 35.0%), and a higher rate of cesarean section (40.4% vs. 17.5%). A total of 36 (34%) of patients with endometriosis had taken medical therapy for endometriosis. Regarding the endometriosis subtype, 77/106 (72.6%) patients presented an ovarian endometriosis only, 13/106 (12.3%)—a DIE only, 10/106 (9.4%)—both ovarian endometriosis and DIE, 4/106 (3.8%)—both ovarian endometriosis and peritoneal endometriosis, and 2/106 (1.9%) presented ovarian endometriosis, DIE, and peritoneal endometriosis.

The analysis of MTHFR polymorphism distribution according to endometriosis diagnosis showed a lower rate of MTHFR wild-type (22.6% vs. 39.9%, *p* = 0.0014) and a higher rate of MTHFR C677T homozygous polymorphism—A1298C wild-type in patients with endometriosis (24.5% vs. 15.8%, *p* = 0.0453) (Table 2).

Table 3 reports the comparison of MTHFR polymorphism distribution according to the presence of ovarian endometriosis, regardless of the presence of concomitant peritoneal endometriosis (n = 81) or DIE endometriosis and regardless of the presence of concomitant ovarian or peritoneal endometriosis (n = 25). No difference in MTHFR polymorphism distribution emerged between those two groups.

The Odds Ratio (OR) for endometriosis diagnosis in patients with MTHFR C677T homozygous polymorphism—A1298C wild-type was 1.7266 (95% CI 1.0068—2.9608, *p* = 0.0472). Binomial logistic regression was performed to ascertain the effects of age, body mass index (BMI), history of Mullerian anomalies, tobacco use, alcohol use, frequent menstrual cycle, heavy menstrual cycle, history of infertility, no previous pregnancy, at least one previous cesarean section, and presence of MTHFR C677T homozygous polymorphism on the likelihood that the included patients had endometriosis. The logistic regression model was statistically significant, χ2(10) = 21.306, *p* = 0.0191. The model correctly classified 74.6% of cases. The variables “tobacco use” and “alcohol use” were not retained in the model. Of the ten included predictor variables, only two were independently associated with endometriosis diagnosis: the presence of an MTHFR C677T homozygous polymorphism and no previous pregnancy (Table 4).

Patients with the MTHFR C677T homozygous polymorphism had 1.889 times higher odds of having endometriosis diagnosis than patients without the MTHFR C677T homozygous polymorphism.

## 3. Discussion

This study showed that, after controlling for confounders, an MTHFR C677T homozygous polymorphism might be considered a risk factor for endometriosis, with an adOR of 1.889. Patients with an endometriosis diagnosis also seem to present a lower prevalence of the MTHFR wild-type (22.6% vs. 39.9%) than those without endometriosis.

The reported results are substantially in line with previous evidence on the topic. Indeed, in 2022, Clément et al. reported a higher prevalence of the C677T homozygous polymorphism (OR 1.74) and a lower prevalence of the MTHFR wild-type (OR 0.42) in 158 infertile women with endometriosis compared to 1430 infertile controls without endometriosis [17]. In 2011, Szczepańska et al. concluded that single nucleotide polymorphisms in genes encoding folate metabolism enzymes may contribute to endometriosis-associated infertility through an epistatic interaction of the rs1801133 of MTHFR (C677T) and rs4244593 of phosphatidylethanolamine N-methyltransferase (PEMT) [18]. On the other hand, Guedes et al. conducted a study on 61 infertile patients with endometriosis and 91 infertile controls in 2022. The authors found no difference in MTHFR allele frequency between cases and controls, and a higher frequency of methionine synthase (MTR gene) G allele and GG genotype, as well as an association at the epistasis analysis of the combination between MTHFR and MTR variants (CC+AG) and pregnancy rate [16]. However, the reported studies focus only on infertile patients with endometriosis and did not account for potential confounders in the association between MTHFR polymorphisms and endometriosis.

The association between an MTHFR C677T homozygous polymorphism and endometriosis could be explained by mechanisms related to epigenetic modifications. Indeed, epigenetic modifications, including DNA hypomethylation, are possible mechanisms by which the expression of several genes necessary for establishing endometriosis is altered [7,8,9]. Increased proliferation, invasion, and resistance to apoptosis may explain the pathogenesis of endometriosis; several of these characteristic behaviors of endometriosis have been previously linked to epigenetic alterations [11].

The presence of an MTHFR C677T homozygous polymorphism may determine a reduction of up to 60–70% of the enzyme activity [19,20], with reduced production of the antioxidant glutathione and a reduced function of the coenzyme S-adenosyl-methionine (SAM), which can determine reduced DNA methyltransferase (DNMT) activity and global DNA hypomethylation [14].

In the literature, the main genes involved in the epigenetic pathway of endometriosis pathogenesis [7,8,9,14] seem to be the estrogen and progesterone receptors ERα (hyper-methylation), Erβ (hypo-methylation) and Prβ (hyper-methylation) [21], the aromatase (development of hyperestrogenic microenvironment) [22], the homebox protein HOXA-10 (hyper-methylation with reduced uterine receptivity), and the Cyclooxygenase COX-2 (over-production of PGE2) [23]. The decreased activity of the antioxidant system caused by the reduced function of the MTHFR may also increase the alterations of the methylation of DNA, b-CpG demethylases, ten-eleven translocation, and jumonji (JMJ), factors involved in the processes of methylation [24]. Estrogen signaling, hypoxia, and inflammation are three interlinked driving forces in the development of endometriosis, and epigenetic components seem to play a central role in coordinating these three factors [9].

The evidenced association between a history of no previous pregnancy and endometriosis was an expected result since it was previously reported as a risk factor for endometriosis [4].

The limitations of this study are related to its retrospective nature since it was not possible to ascertain the effect of unmeasured confounding factors on the outcome, like dietary habits [25], family history of endometriosis [26], and environmental factors [27] or for the exposure, like family history of MTHFR polymorphisms. The prevalence of endometriosis in our population (25.9%) seems to be higher than the prevalence in the general population. Still, it is comparable to those reported for the selected population attending outpatient clinics or who underwent medical or surgical intervention (23.8–49.7%) [28,29]. The available data made it not possible to determine the indications for MTHFR testing. It is reported that MTHFR testing is among the most ordered genetic testing (fourth in Italy), with hyperhomocysteinemia, venous thrombotic event, cerebrovascular accident, family history, intrauterine fetal demise, recurrent miscarriage, lupus, vasculitis, transplant evaluation, immune thrombocytopenic purpura, migraines, and depression reported as indications for testing, with lack of uniformity and low evidence-based recommendations [30,31]. However, the lack of standardization of indications for MTHFR polymorphism testing has not likely determined a selection bias for this study, since having performed an MTHFR polymorphism test did not represent a risk factor for the study outcome. Our results are strengthened by having included a large number of patients, with different subtypes of endometriosis, both with and without endometriosis-related infertility, and with an a priori determined sample size, allowing us to run a logistic regression with a significant number of confounders. Moreover, we only included patients with results of MTHFR polymorphisms tests from the same laboratory and histopathological diagnosis of endometriosis, thus lowering the risk of recall bias.

The results of this study should be interpreted with caution before being able to draw conclusions to be reported in clinical practice. The highlighted association between MTHFR C677T homozygous polymorphism and endometriosis should be, at the moment, considered a preliminary guide for further studies. Indeed, efforts should be made to perform molecular analysis on tissues from patients with endometriosis and MTHFR C677T homozygous polymorphism to evaluate the methylation of genes implicated in endometriosis pathogenesis and consider the antioxidant activity, the SAM function, the folate levels, and dietary habits.

## 4. Materials and Methods

This was a retrospective case–control study conducted on all patients attending the gynecological outpatient clinic of our institution (Clinica di Ostetricia e Ginecologia, Azienda Ospedaliero-Universitaria delle Marche, Università Politecnica delle Marche, Ancona, Italy) from January 2020 to December 2022, who had previously performed an MTHFR polymorphisms test (for both C677T and A1298C polymorphisms) in the laboratory of our institution, regardless of the indication. Among these patients, we defined only those who had undergone surgical procedures in the past with an available written histopathological diagnosis of endometriosis as cases and those who had never received a histopathological diagnosis of endometriosis, without signs of endometriosis in the gynecological examination or in the pelvic transvaginal ultrasound, and without endometriosis-related symptoms as controls. Exclusion criteria were age < 18 years, menopause, and data unavailability or incompleteness. The research was conducted according to the Declaration of Helsinki. According to Italian legislation, the local ethical committee of our institution (Comitato Etico Regionale Marche) approved the study protocol (n° CERM 2023/91).

Patients attended our gynecological outpatient clinic in cases of pelvic pain, abnormal uterine bleeding, vaginal discharge, vulvar itching or swelling, incontinence, pelvic floor disorders, pre-conception counseling, or follow-up of previously diagnosed gynecological conditions. According to our local protocols, after obtaining informed consent for clinical procedures and data collection for clinical and research purposes, all patients underwent history taking, gynecological examination, and a pelvic transvaginal/transabdominal ultrasound. Ultrasounds were performed by gynecologists with particular expertise in gynecological ultrasound, with a 3.5–5.5 mHz probe, a Voluson E10 (GE Healthcare, Milwaukee, WI, USA), and a standardized approach consisting of transvaginal evaluations of the uterus and adnexa (uterus mobility, possible presence of adenomyosis, or ovarian cysts), of the possible presence of site-specific tenderness, of ovarian mobility, of the possible pouch of Douglas obliteration by the sliding sign, of the possible presence of deeply infiltrative endometriosis (DIE) (hypoechoic irregular-shaped nodules with a hyperechoic rim) in the anterior compartment (bladder, uterovesical region, and ureter) or posterior compartment (rectovaginal septum, posterior and lateral vaginal fornix, uterosacral ligaments, anterior rectum/anterior rectosigmoid junction, and sigmoid colon), and pain mapping. A transabdominal scan was also performed to evaluate the kidneys and the presence of hydronephrosis. When appropriate, the ultrasound examination was completed with transrectal sonography [32,33,34,35,36].

For this study, all data were retrospectively retrieved from outpatient clinical charts and entered into a dedicated database with a unique study ID for each patient. Patients with missing or incomplete data were excluded from the analysis. The following variables were collected: age (years); BMI (kg/m^2^); history of Mullerian anomalies; tobacco use; alcohol use; menstrual cycle frequency: absent, infrequent (>38 days), normal (≥24 to ≤38 days), frequent (<24 days) [37]; menstrual cycle flow volume: light, normal, or heavy [37]; history of infertility; the number of previous pregnancies; the outcome of the previous pregnancies (vaginal birth, cesarean section, spontaneous miscarriage, or ectopic pregnancy); rate of recurrent miscarriage (≥three previous spontaneous miscarriages); medication history; MTHFR polymorphisms status (C677T and A1298C wild-type, C677T homozygous and A1298C wild-type, C677T heterozygous and A1298C wild-type, C677T wild-type and A1298C homozygous, C677T wild-type and A1298C heterozygous, C677T heterozygous and A1298C heterozygous, C677T homozygous and A1298C heterozygous, C677T heterozygous and A1298C homozygous, or C677T homozygous and A1298C homozygous); and homocysteinemia (μmol/L). In cases of endometriosis diagnosis, we also reported the endometriosis subtype according to the surgical report (ovarian endometriosis, DIE, or peritoneal endometriosis). The outcome of this study was the diagnosis of endometriosis. We considered the presence of an MTHFR C677T homozygous polymorphism as exposure. The MTHFR C677T homozygous polymorphism was chosen as the exposure since it is reported that it is associated with a higher reduction in enzyme function (up to 60–70%) than other MTHFR polymorphisms [19,20]. We considered confounders all the variables that are reported to be risk factors for endometriosis diagnosis: age, BMI, history of Mullerian anomalies, tobacco use, alcohol use, frequent menstrual cycle, heavy menstrual cycle, history of infertility, no previous pregnancy, or at least one previous cesarean section [4,26,38,39,40,41,42].

### Statistical Analysis

The G*Power version 3.1.9 software was used to determine the required sample size using the method described by Hsieh et al. for logistic regression with a binomial outcome (dependent variable: endometriosis/no endometriosis) [43]. According to previous literature, the prevalence of MTHFR C677T homozygous polymorphism in the European general population is 13.5% (H0 = 0.135) [44]. We assumed that the prevalence of MTHFR C677T homozygous polymorphism should double in endometriosis-affected patients to be considered significantly different from the general population (H1 = 0.27). Considering α = 0.05, a power of 0.80, an R2 other than X of 0.3 for the confounders (moderate association), and an X parm Π of 0.5, the total sample size resulted in 402 patients. We included patients from January 2020 to December 2022 to reach the required sample size. Statistical software SPSS 28.0.1.1 (SPSS Inc., Chicago, IL, USA) was used for data analysis. All continuous variables were tested for normality with the D’Agostino–Pearson test. Normally distributed variables were expressed as mean ± standard deviation (SD), while not-normally distributed variables were reported as median and interquartile range (IQR). The *t*-test or the Mann–Whitney test was used for comparison as appropriate. Qualitative variables were expressed as number and proportions and were compared with the Chi-square test or the Fisher’s exact test as appropriate. All the collected background and clinical variables were compared between cases and controls in a bivariate analysis. The odds ratio (OR) with a 95% confidence interval (CI) for endometriosis diagnosis was determined according to MTHFR C677T homozygous polymorphism status. None of the considered variables had the characteristics of an instrumental variable [45]. A logistic regression with a binomial dependent variable (endometriosis/no endometriosis) was run, and the adjusted OR for endometriosis diagnosis according to MTHFR C677T homozygous polymorphism (exposure) status was determined. We included the following variables as covariates: age, BMI, history of Mullerian anomalies, tobacco use, alcohol use, frequent menstrual cycle, heavy menstrual cycle, history of infertility, no previous pregnancy, and at least one previous cesarean section. A *p* < 0.05 was considered statistically significant.

## 5. Conclusions

MTHFR C677T homozygous polymorphism could be considered a risk factor for endometriosis, after controlling for confounders. Epigenetic modifications may be the most important mechanism explaining the observed association, through the processes of altered DNA methylation and reduced activity of antioxidant systems.

## Figures and Tables

**Figure 1 ijms-24-15404-f001:**
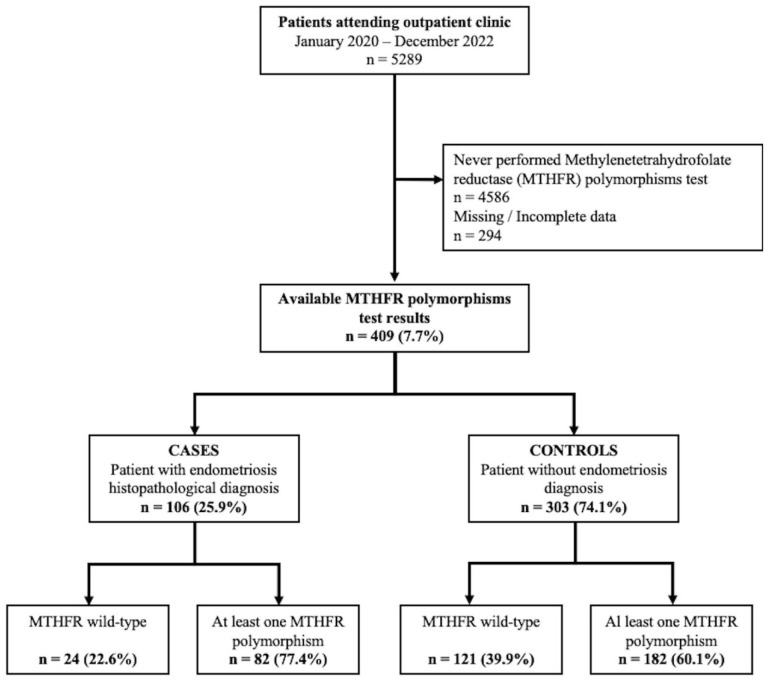
Flow-chart of the study population.

**Table 1 ijms-24-15404-t001:** Background and clinical variables of the patients included, according to endometriosis diagnosis.

Variable	Cases n = 106	Controls n = 303	*p* *
Age (years)	41.7 ± 9.0	43.6 ± 11.3	0.1035
Body Mass Index (BMI)	24.4 ± 4.2	24.9 ± 4.1	0.2321
History of Mullerian anomalies	4 (3.8)	16 (5.3)	0.5390
Tobacco use	3 (2.8)	6 (2.0)	0.6299
Alcohol use	1 (0.9)	0 (0.0)	0.2591
Frequent menstrual cycle	7 (6.6)	26 (8.5)	0.5355
Heavy menstrual cycle	16 (15.1)	53 (17.4)	0.5861
History of infertility	16 (15.1)	34 (11.2)	0.2918
N° previous pregnancies	0 (0–1)	1 (0–2)	0.0002
No previous pregnancy	54 (50.9)	106 (35.0)	0.0039
At least one spontaneous miscarriage ^	10 (19.2)	59 (29.9)	0.1257
Recurrent miscarriage ^	0 (0.0)	16 (2.7)	0.2319
At least one cesarean section ^	21 (40.4)	53 (17.5)	0.0004
Homocysteinemia (μmol/L)	11.5 ± 5.6	11.9 ± 4.4	0.5013

Data are reported as mean ± SD, median (IQR), or n (%) as appropriate. * *t*-test, Mann–Whitney Test, chi-squared test, or Fisher’s exact test as appropriate. ^ Only patients with at least one previous pregnancy (n = 52, and n = 197, respectively).

**Table 2 ijms-24-15404-t002:** MTHFR polymorphism distribution according to endometriosis diagnosis.

MTHFR Status		Cases n = 106	Controls n = 303	*p* *
C677T	A1298C			
wild-type	wild-type	24 (22.6)	121 (39.9)	0.0014
homozygous	wild-type	26 (24.5)	48 (15.8)	0.0453
heterozygous	wild-type	25 (23.6)	64 (21.1)	0.5917
wild-type	homozygous	7 (6.6)	13 (4.3)	0.3455
wild-type	heterozygous	11 (10.4)	18 (5.9)	0.1201
heterozygous	heterozygous	12 (11.3)	37 (12.2)	0.8061
homozygous	heterozygous	0 (0.0)	2 (0.7)	0.3884
heterozygous	homozygous	0 (0.0)	0 (0.0)	-
homozygous	homozygous	1 (0.9)	0 (0.0)	0.0987

Data are reported as n (%). * chi-squared test or Fisher’s exact test as appropriate (total chi-squared: *p* = 0.0190).

**Table 3 ijms-24-15404-t003:** MTHFR polymorphism distribution according to endometriosis subtype.

MTHFR Status		Ovarian Endometriosis n = 81	Deeply Infiltrative Endometriosis n = 25	*p* *
C677T	A1298C			
wild-type	wild-type	17 (21.1)	7 (28.0)	0.6462
homozygous	wild-type	21 (25.9)	5 (20.0)	0.7368
heterozygous	wild-type	21 (25.9)	4 (16.0)	0.4518
wild-type	homozygous	4 (4.9)	3 (12.0)	0.4341
wild-type	heterozygous	9 (11.1)	2 (8.0)	0.9436
heterozygous	heterozygous	9 (11.1)	3 (12.0)	0.8116
homozygous	heterozygous	0 (0.0)	0 (0.0)	-
heterozygous	homozygous	0 (0.0)	0 (0.0)	-
homozygous	homozygous	0 (0.0)	1 (4.0)	0.5319

* Chi-square test or Fisher’s exact test as appropriate.

**Table 4 ijms-24-15404-t004:** Logistic regression for endometriosis diagnosis.

Variable	adOR	95% CI	B	Std. Error	Wald	*p*
Age	0.985	0.963–1.008	−0.015	0.012	1.655	0.1983
BMI	0.964	0.910–1.020	−0.037	0.029	1.624	0.2026
Mullerian anomalies	0.541	0.169–1.732	−0.614	0.593	1.070	0.3009
Frequent menstrual cycle	0.835	0.339–2.057	−0.180	0.460	0.153	0.6956
Heavy menstrual cycle	0.853	0.450–1.619	−0.160	0.326	0.239	0.6248
History of infertility	1.398	0.724–2.700	0.335	0.336	0.993	0.3189
No previous pregnancy	2.191	1.295–3.708	0.784	0.268	8.538	0.0035
1+ cesarean section	1.750	0.916–3.346	0.560	0.331	2.867	0.0904
MTHFR C677T homozygous	1.889	1.076–3.318	0.636	0.287	4.899	0.0269
Homocysteinemia	0.985	0.939–1.033	−0.015	0.025	0.396	0.5291
Constant	0.985	0.963–1.008	−0.015	0.012	1.655	0.1983

Variables not included in the model: tobacco use and alcohol use.

## Data Availability

Data are available from the corresponding author upon reasonable request.

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
