# Peer review of "Homozygous C677T Methylenetetrahydrofolate Reductase (MTHFR) Polymorphism as a Risk Factor for Endometriosis: A Retrospective Case–Control Study"

_ijms, 2023, doi:10.3390/ijms242015404_

Round 1

Reviewer 1 Report

The aim of this investigation has been to identify a new genetic risk factor for endometriosis. In my opinion, it contains useful information, but – before it can be accepted – there are preliminary issues/questions that need to be clarified/answered.

For this reason, I will not report specific, minor issues.

GENERAL ISSUES

1.            The manuscript has an unusual layout: Materials and Methods are described after the Discussion. Instead, they should come immediately after the Introduction, so that the reader knows how the study has been conducted before getting to the results.

2.            Determining the methylenetetrahydrofolate (MTHFR) gene polymorphism is definitely not a routine investigation, especially in an outpatient gynecological clinic. Under the circumstances, Authors should explain why in their clinic (or, for that matter, in their hospital) such a sophisticated test has been routinely performed. It sounds as if this happened as part of a separate investigation of eating habits of subjects attending their hospital. If so, it should be mentioned.

3.            The validity of the investigation depends entirely on the accuracy of the diagnosis of endometriosis. From the description in Materials and Methods, it appears that the presence or absence of endometriosis was determined through “pelvic transvaginal/transabdominal ultrasound”. Authors specify that the ultrasound evaluation was performed by “by gynecologists with particular expertise in gynecological ultrasound, with a 3.5-5.5 mHz probe, a Voluson E10 (GE Healthcare, Milwaukee, WI, USA), and a standardized approach…”. This poses a problem: as recently reinforced (Chen-Dixon K, Uzuner C, Mak J, Condous G. Effectiveness of ultrasound for endometriosis diagnosis. Curr Opin Obstet Gynecol. 2022; 34:324-31), “Although ultrasound can detect adenomyosis, deep endometriosis and endometriomas, it is not possible to reliably detect superficial endometriosis”. The text mentions that “A histopathological diagnosis of endometriosis was reported in 106 (25.9%) patients (cases). The remaining 303 (74.1%) never had a histopathological diagnosis of endometriosis, and endometriosis was excluded at the gynecological examination with pelvic transvaginal ultrasound”. Since superficial peritoneal endometriosis is the most common form of the disease, the validity of this investigation is subject to question. In my opinion, before this report can be accepted and evaluated in detail, this fundamental question must be resolved.

4.            The statistical evaluation of the data should be checked by a statistician, since I am not able to do so.

No problem detected

Author Response

Prof. Andrea Ciavattini

Università Politecnica delle Marche

Via F. Corridoni, 11 60123 Ancona - Italy

Tel: +39 071 36745 Fax: +39 071 36575

e-mail: ciavattini.a@libero.it

Ancona, 09-October-2023

Ref: Manuscript ijms-2622357 “Homozygous C677T methylenetetrahydrofolate reductase (MTHFR) polymorphism as a risk factor for endometriosis: a retrospective case-control study”

Dear Beiner Zhang,

Editor

Dear Reviewers,

Thank you for your positive response to our journal submission. We appreciate the constructive feedback; we have studied your comments carefully and made some corrections, which we hope to meet with your approval.

We added changes in the manuscript and marked them with red text to find the changes reading the paper.

Best regards,

Andrea Ciavattini

Reviewer 1

The aim of this investigation has been to identify a new genetic risk factor for endometriosis. In my opinion, it contains useful information, but – before it can be accepted – there are preliminary issues/questions that need to be clarified/answered.

For this reason, I will not report specific, minor issues.

GENERAL ISSUES

  1. The manuscript has an unusual layout: Materials and Methods are described after the Discussion. Instead, they should come immediately after the Introduction, so that the reader knows how the study has been conducted before getting to the results.

            We prepared the manuscript according to the IJMS Instructions for Authors, available at https://www.mdpi.com/journal/ijms/instructions, which specifically indicate that the Results section should immediately follow the Introduction (Research manuscript sections: Introduction, Results, Discussion, Materials and Methods, Conclusions (optional).

  1. Determining the methylenetetrahydrofolate (MTHFR) gene polymorphism is definitely not a routine investigation, especially in an outpatient gynecological clinic. Under the circumstances, Authors should explain why in their clinic (or, for that matter, in their hospital) such a sophisticated test has been routinely performed. It sounds as if this happened as part of a separate investigation of eating habits of subjects attending their hospital. If so, it should be mentioned.

            Thank you for this comment; indeed, MTHFR polymorphisms test was not routinely performed in the included patients. The patients underwent the test in the past for various indications and not for the diagnosis of endometriosis. In order to conduct this retrospective study, we cross-referenced data from the databases of our laboratory and our outpatient clinic to identify patients who had performed the MTHFR test in the past and who had had a gynecologic evaluation at our clinic. We clarified those concepts in the Materials and Methods and Results section (see page 2, line 60, page 8 line 277 and line 279).

  1. The validity of the investigation depends entirely on the accuracy of the diagnosis of endometriosis. From the description in Materials and Methods, it appears that the presence or absence of endometriosis was determined through “pelvic transvaginal/transabdominal ultrasound”. Authors specify that the ultrasound evaluation was performed by “by gynecologists with particular expertise in gynecological ultrasound, with a 3.5-5.5 mHz probe, a Voluson E10 (GE Healthcare, Milwaukee, WI, USA), and a standardized approach…”. This poses a problem: as recently reinforced (Chen-Dixon K, Uzuner C, Mak J, Condous G. Effectiveness of ultrasound for endometriosis diagnosis. Curr Opin Obstet Gynecol. 2022; 34:324-31), “Although ultrasound can detect adenomyosis, deep endometriosis and endometriomas, it is not possible to reliably detect superficial endometriosis”. The text mentions that “A histopathological diagnosis of endometriosis was reported in 106 (25.9%) patients (cases). The remaining 303 (74.1%) never had a histopathological diagnosis of endometriosis, and endometriosis was excluded at the gynecological examination with pelvic transvaginal ultrasound”. Since superficial peritoneal endometriosis is the most common form of the disease, the validity of this investigation is subject to question. In my opinion, before this report can be accepted and evaluated in detail, this fundamental question must be resolved.

            Thank you for this comment. We considered as cases only those patients with a histopathological diagnosis of endometriosis obtained from previous surgical procedures. The inclusion as case was not related to the ultrasound evidence of endometriosis. In order to be considered as a Control, a patient should have never received a histopathological diagnosis of endometriosis and have no sign of endometriosis on the gynecological examination or on the pelvic transvaginal ultrasound. We better clarified those selection criteria in the Materials and Methods and Results section (see page 8, line 280-284). We agree that ultrasound may be limited in the identification of peritoneal endometriosis, but we believe that this condition did not significantly affect our results, since the prevalence of peritoneal endometriosis alone in asymptomatic patients is estimated to be of 2% [Damario, M.A., Rock, J.A. (2000). Peritoneal Endometriosis. In: diZerega, G.S. (eds) Peritoneal Surgery. Springer, New York, NY. https://doi.org/10.1007/978-1-4612-1194-5_22]. We added the clarification that patients included in the Control group were asymptomatic for endometriosis-related symptoms (see page 8, line 284). 

The statistical evaluation of the data should be checked by a statistician, since I am not able to do so.

Reviewer 2

Dear authors, I have read your article with interest. Here are my suggestions and remarks:

- you wrote that primary outcome was diagnosis of endometriosis - is that correct?

            We confirm that the primary outcome was a diagnosis of endometriosis. Since it was the main study outcome (i.e., we evaluated the effect of an exposure, the presence of an MTHFR C677T homozygous polymorphism, on an outcome, an endometriosis diagnosis), we removed the term “primary” to avoid misunderstandings.

- were there any secondary outcomes?

            No, the only study outcome was the diagnosis of endometriosis (see the answer above).

- how many patients had histological diagnosis of endometriosis and in how many diagnosis was made only by ultrasound ?

Thank you for this request of clarification. All patients included in the study group had a histopathological diagnosis of endometriosis obtained from previous surgical procedures. The inclusion as case was not related to the ultrasound evidence of endometriosis. In order to be considered as a Control, a patient should have never received a histopathological diagnosis of endometriosis and have no sign of endometriosis on the gynecological examination or on the pelvic transvaginal ultrasound. We better clarified those selection criteria in the Materials and Methods and Results section (see page 8, line 280-284).

- do you have data on treatment of endometriosis (surgical, with medication?)

Thank you for this comment. All included patients with endometriosis underwent surgical treatment. We added the specification regarding how many patients had taken a medical therapy (see page 4, line 137-138).

- are there any differences in polymorphisms between ovarian endometriosis and DIE?

Thank you for this comment. We added the comparison of MTHFR polymorphisms distribution according to the presence of ovarian endometriosis or DIE endometriosis. No difference in MTHFR polymorphisms distribution emerged between those two groups (see page 4, line 138-142, and Table 3).

- in the discussion you should focus mainly on your results. Other mechanisms can be mentioned but there is no need to discuss them in detail.

Thank you for this comment. We summarized the part of the discussion regarding potential pathogenetic mechanisms (see page 7, line 228-233). 

Moreover the discussion section does not thoroughly interpret the study's findings in the context of existing literature. It should discuss how the results contribute to the current understanding of endometriosis and MTHFR polymorphisms.

- there are articles about homo and heterozygous state of mutations and their effect on endometriosis. Any comment on that in the discussion?

Thank you for those two comments. We expanded the Discussion section reporting the comparison or our results with other studies on the topic (see page 6, line 199-214). 

- what are the strenghts of your study. what does the study add to the already known facts in the literature (connection between MTHFR and endometriosis has already been widely discussed)

            Thank you for this comment. Unlike previous studies, we did not focus only on infertile patients, and we included patients with different subtypes of endometriosis. Moreover, we evaluated the association between MTHFR polymorphisms and endometriosis also considering the potential effect as confounders of known risk factors for endometriosis, thus lowering the risk of bias. We better clarify those concepts in the Discussion section (see page 7, line 258-261 and age 8, line 262-264). 

Reviewer 2 Report

Dear authors, I have read your article with interest. Here are my suggestions and remarks:

- you wrote that primary outcome was diagnosis of endometriosis - is that correct?

- were there any secondary outcomes?

- how many patients had histological diagnosis of endometriosis and in how many diagnosis was made only by ultrasound ?

- do you have data on treatment of endometriosis (surgical, with medication?)

- are there any differences in polymorphisms between ovarian endometriosis and DIE?

- in the discussion you should focus mainly on your results. Other mechanisms can be mentioned but there is no need to discuss them in detail. Moreover the discussion section does not thoroughly interpret the study's findings in the context of existing literature. It should discuss how the results contribute to the current understanding of endometriosis and MTHFR polymorphisms.

- there are articles about homo and heterozygous state of mutations and their effect on endometriosis. Any comment on that in the discussion?

- what are the strenghts of your study. what does the study add to the already known facts in the literature (connection between MTHFR and endometriosis has already been widely discussed)

Author Response

(The authors gave the same response as above.)

Round 2

Reviewer 1 Report

IN my opinion the Authors have replied satisfactorily to the observations/queries by the Reviewers.

Therefore, this article is now ready for publication

Reviewer 2 Report

Thank you for your answers and comments to my questions. I believe the answers are satisfactory. I have no additional comments.